# Spatiotemporal Characteristics of Fish Larvae and Juveniles in the Waters around Taiwan from 2007 to 2019

**DOI:** 10.3390/ani12151890

**Published:** 2022-07-25

**Authors:** Kuo-Wei Yen, Chia-I Pan, Chia-Hsiang Chen, Wei-Hsiang Lien

**Affiliations:** Fisheries Division, Taiwan Fisheries Research Institute, Council of Agriculture, Keelung 202008, Taiwan; kwyen@mail.tfrin.gov.tw (K.-W.Y.); cjpan@mail.tfrin.gov.tw (C.-I.P.); chchen.chen@gmail.com (C.-H.C.)

**Keywords:** climate change, fishery, ORI net, early life history, ecosystem dynamics, extreme rainfall, Morakot

## Abstract

**Simple Summary:**

Fish larvae and juveniles are necessary fishery recruitment resources. As climate change and natural climate events continue to impact marine ecology, it may become difficult to determine the characteristics of changes in fish larvae and juveniles. Using samples and data from long-term marine experimental monitoring, we found a high diversity of fish larvae and juveniles in the waters around Taiwan, and the abundance of different fish species varied spatially and seasonally. We also found that distance from the coastline and topography were the key factors affecting the community of fish larvae and juveniles. By presenting these data as times series, we confirmed that 2009 was a critical year for regime change between fish larvae and juveniles in different depth zones. The year also happened to include Taiwan’s worst typhoon on record. These results emphasize the need to conduct more detailed research to prevent predictable and unpredictable shocks.

**Abstract:**

Taiwan is located at the intersection of tropical and subtropical islands in the western Pacific Ocean. This area is an important spawning and breeding ground for many economic and noneconomic species; however, little is known about the long-term dynamics of fish larvae and juveniles in these waters. In this study, we conducted an in-depth exploration of their spatial characteristics using 2007–2019 field survey samples. Our results demonstrated the seasonality and spatiality of the larvae and juveniles of different fish species. We also found that the continental shelf and offshore distance were key factors affecting fish larvae and juveniles. Changes in community structure were temporally correlated with the extreme rainfall of Typhoon Morakot (the worst typhoon ever recorded in Taiwan). These data can be used as a management reference for fisheries’ policymaking and provide key insights into nearby marine ecosystems and the early life history of fish.

## 1. Introduction

Understanding the early life history of fish is vital to resource research because this period is a crucial determinant before resources are recruited to the fishery [1]. When early life history is affected by environmental changes, fluctuations in fishery resources may result [2,3] and lead to future additions to fish stocks [4,5]. Changes in environmental factors, such as temperature, salinity, the abundance of prey organisms, competition, and ocean current transport, affect the abundance and distribution of fish larvae and juveniles [6,7,8]. Therefore, it is highly important for fishery resource management to conduct related research on fish larvae and juveniles in the surrounding waters.

Previous studies performed in the Taiwan Strait explored the impact of the monsoon system on aggregations of fish larvae and juveniles [9,10,11]. For example, Hsieh et al. [12] discussed the composition and distribution of fish larvae and juveniles in the waters around Taiwan in winter. Chen et al. [13] analyzed the composition of fish larvae and juveniles in the East China Sea and the waters around Taiwan in winter, and Wang et al. [14] conducted a related study on fish larvae and juveniles and the hydrological environment in the invasion area of the Kuroshio Current in the southern East China Sea, northern Taiwan.

The Kuroshio Current is the main route of spawning migration for many species [15]. In addition, the mixing of this current with the areas along boundary currents produces higher-quality habitats for organisms; thus, the waters surrounding Taiwan are a vital fishing ground and one of the most biodiverse regions worldwide [16]. In high-biodiversity regions, environmental changes and changes in the structure of biological communities are the key factors affecting changes in organismal abundance [17]. Changes in prey organisms will affect the growth and survival of fish in their early life stages [17,18]. For example, zooplankton are essential prey in the early life stage of fish. Studies have shown that a change in the plankton community with increasing water temperature will reduce the survival probability of economic fish species [19,20]. When the amount of prey is insufficient to support fish growth, the growth rate of fish will slow [21,22]. The Kuroshio region and its surrounding waters are home to surface fishes [15,23,24], mesopelagic fishes [25,26], and ground-dwelling fishes [27], and these areas are essential spawning and nursery grounds. Most fish exhibit a planktonic period in the early life stage, as this characteristic ensures that a species has sufficient prey and can spread to more areas, increasing the chances of survival and population sustainability [6]. In the relatively warm and nutrient-poor waters of the Kuroshio area, environmental changes may impact the entire ecosystem as well as the production chain of fisheries and thereby severely impact aquatic production in neighboring countries.

Many studies have shown that as global warming intensifies, the Kuroshio Current is accelerating the warming process [28,29]. In recent years, studies have revealed that climate change affects ocean current intensity [30]. The Kuroshio Current is strongly affected by short-term climatic events (e.g., the Pacific Decadal Oscillation (PDO) or the El Niño–Southern Oscillation (ENSO)) [31,32,33,34,35]; these phenomena and warming lead to changes in the path of the current [36], which in turn affects zooplankton communities and the entire ecosystem [37,38]. Previous studies have shown that changes in the aggregation and distribution of fish larvae and juveniles are strategies that alter spawning and habitat selection under various conditions [39,40,41]. The depth and stratified thermohaline properties of seawater determine the time and location of spawning [42]. Additionally, fish foraging and production conditions will change with alterations in the marine environment [43]. Brood stocks will select a suitable spawning ground with low mortality of fish larvae and juveniles [44]. Some changes in the physical environment along the coast and in inshore regions have been confirmed to affect fish larvae and juveniles. For example, ocean currents, rivers, oceans, tides, or terrain can result in internal waves or upwelling that alters characteristics such as larval swarming and growth; thus, these processes further affect the distribution of fish larvae and juveniles [13,45,46].

There is considerable uncertainty in estimating the dynamics of fishery resources based on short-term or limited-year data. However, past studies and investigations of fish larvae and juveniles in Taiwan considered sampling stations in different locations mainly due to different voyages or were restricted by the lack of long-term observational data. As a result, it has been impossible to systematically describe the abundance and distribution of seasonal changes in larvae and juveniles of important economic fish in the surrounding waters.

The Taiwan Fisheries Research Institute built the 1948-ton Fishery Researcher Vessel 1 and 345-ton Fishery Researcher Vessel 2 in 1993 and 2013, respectively. The two vessels are used to investigate fishing environments and fishery resources in coastal areas. To establish long-term and systematic marine environmental observation datasets to understand the temporal and spatial distributions of fish larvae and juveniles in the waters around Taiwan, this study carried out 19 voyages in the surrounding waters from 2007 to 2019 using the two fishery research vessels. Samples of fish larvae and juveniles from the surrounding waters were collected from 62 stations to explore their ecological distribution and the main marine environmental factors affecting changes in abundance; additionally, these samples were used to understand their early life history characteristics and thus serve as a reference for fishery resource assessment and management. The specific goals of this research were as follows:To identify the occurrence probability of primary fish larvae and juveniles in different regions and seasons in the waters around Taiwan.To identify the main population structure divisions of important fish larvae and juveniles in Taiwan’s waters.To explore the temporal and spatial composition of fish larvae and juveniles in the pelagic, mesopelagic, and demersal zones.

## 2. Materials and Methods

### 2.1. Data Collection

For this study, two research vessels, Fishery Researcher Vessel 1 and Fishery Researcher Vessel 2 of the Taiwan Fisheries Research Institute, conducted 19 voyages from 2007 to 2019 to collect fish larval and juvenile samples and hydrological data from 62 stations. According to the month of operation, each collected sample was categorized as occurring in one of the four seasons: March, April, and May were categorized as spring; June, July, and August were categorized as summer; September, October, and November were categorized as fall; and December, January, and February were categorized as winter. The depth distribution partitioning scheme is shown in Figure 1. The actual experimental location deviates from each station’s raw latitude and longitude because the research vessel continued to move during the experiment. Therefore, zooplankton collection was performed first at each station, and care was taken to minimize spatial differences in the collections. For the convenience of analysis, we assumed that the fish larvae and juveniles at the experimental site are the same as those at the original station and use the raw latitude and longitude for each collection. An Ocean Research Institute (ORI) net [47] was used as a plankton net (160 cm in diameter, 6 m in length, with 330-μm mesh) and trawled diagonally upward from the target depth. The target depth of the net was set at 200 m. The depth of the ORI net was controlled according to the speed of the vessel and the angle of the steel wire attached to the net, and both factors were constantly monitored and adjusted. If the water depth of the station was less than 200 m, the actual depth was determined to be 5 m above the maximum depth; after the depth was set, the net was towed obliquely from the desired depth toward the water surface while the boat moved at a speed of 1 m/s. For safety reasons, we required a certain speed during actual operation to prevent the net from being pulled into the bottom of the boat. While the net was in the water, we constantly checked that the angle of the wire was maintained at 45 degrees to ensure the correct depth of the net. A one-way clutch mechanical flowmeter (Model 2030RC, General Oceanics Inc., Miami, FL, USA) was attached to the mesh port to calculate the total filtered water volume of the mesh for each sampling. The obtained samples were washed repeatedly with seawater to flush organisms into a bottle placed at the bottom of the net for collection; then, the biological samples were put into the sample bottle. The samples were preserved in a 5% formalin-seawater solution and taken to the laboratory for biomass determination and classification.

### 2.2. Classification and Identification

The zooplankton samples collected with the plankton net were brought back to the laboratory and then uniformly and randomly divided into two subsamples with a Folsom Plankton Sample Splitter [48]; all the fish larvae and juveniles in one of the subsamples were selected and placed under a dissecting microscope for classification and counting. The identification of fish larvae and juveniles were mainly identified according to the works of Mododa [48], Okiyama [49], and Chiu [50]. We also confirmed the growth stage of the fish body on the basis of Leis and Rennis [51] and Kendall Jr. [52]. The fish larval and juvenile abundance (ind./1000 m^3^) was converted by multiplying the number of individuals in the subsample by two and dividing by the filtered water volume of the plankton net. The depth zones of different fish species are listed according to the classification methods of Sassa and Konishi [25] and Chen, Chen, Wang and Lee [13]. A total of 38 species of larval and juvenile fish are frequently caught in the waters around Taiwan, and these species accounted for 70% of the samples collected in this study. The different depth zones and codes are detailed in Table 1.

### 2.3. Analysis of the Occurrence Probability of Fish Larvae and Juveniles at the Species Level

In this study, according to different seabed topographies and regions, the waters around Taiwan were divided into the northwest, northeast, southeast, south, southwest, and Penghu regions (Figure 1). Fish larvae and juveniles were classified according to the geography of the catch areas, and seasons were classified according to catch time. The occurrence probability of each species in a given depth zone was calculated as follows:$$P_{ij}=\frac{N_{ij}}{N_{i}}$$
where $P_{ij}$ represents the probability of catching fish *i*, *j* is one of the six areas defined as previously mentioned, $N_{i}$ represents the total catch of fish *i*, and $N_{ij}$ represents the catch of fish *i* in area *j*. Additionally, the occurrence probability of each species in a given season was calculated as follows:$$P_{ik}=\frac{N_{ik}}{N_{i}}$$
where $P_{ik}$ represents the probability of catching fish *i* in season *k*, $N_{i}$ represents the total catch of fish *i*, and $N_{ik}$ represents the catch of fish *i* in season *k*.

### 2.4. Cluster Analysis

Based on previous studies on the importance of economic fish species reproducing in the waters around Taiwan in spring [53], spring was selected as the season for which to conduct the cluster analysis of the composition of fish larvae and juveniles. Statistical programming was performed using the R statistical program version x64 4.1.1 [54]. The analysis, as mentioned above, uses k-means clustering on the mean catch of each species for each season [55]; this method calculates the similarity between fish larvae and juveniles based on the compositional characteristics of each station and then groups them accordingly. In addition, because the grouping method must specify the number of groups ahead of time, the elbow method [56] was used to sequentially calculate the sum of the variation in all groups from 1 to n and plot the XY distribution line graph with the sum of the variation and the number of clusters. The turning point where the XY distribution line exhibited significant variation, which was considered the optimal number of clusters, was identified by the cluster command in the “Cluster” package [57] in the R statistical program.

### 2.5. Analysis of Temporal and Spatial Compositional Changes in Fish Larvae and Juveniles in Different Habitat Depth Zones

Temporal and spatial data on fish larvae and juveniles to calculate their temporal and spatial composition variation characteristics. The formula for the analysis of the spatial variation characteristics is as follows:$$P_{xyz}=\frac{\sum N_{xyz}}{\;\sum FWV_{xy}}\times 2\times 1000$$
where $P_{xyz}$ represents the average probability of occurrence in the space where longitude equals *x*, latitude equals *y*, and depth zone equals *z*; $\sum FWV_{xy}$ represents the total filtered water volume in this space during the study period in m^3^; and $\sum N_{xyz}$ represents the depth zone of the species. We multiplied by two to restore the segmentation of zooplankton and multiplied by 1000 to unify these data with the fish larval and juvenile abundance (ind./1000 m^3^).

The formula for the analysis of time series characteristics is as follows:$$P_{xyt}=\frac{\sum N_{xyt}}{\;\sum FWV_{xy}}\times 2\times 1000$$

The algebraic concept is consistent with the previous formula, where *xy* represents the space and is modified to *t*, which represents the year.

## 3. Results

Photographs of the top three catches of fish species (larvae and juveniles) from three different depth zones (pelagic, mesopelagic, and demersal) in this study are shown in Figure 2. Differences in the occurrence probability of fish larvae and juveniles according to geographic region and season were observed. By analyzing the probability of occurrence of fish larvae and juveniles in different geographic regions over time, we found that individual fish species had different spatial distributions and that most species were present in all sea areas. A small number of fish species frequently appeared in specific sea areas. For example, larvae and juveniles of the mullet (*Mugil cephalus*), which spawns in southwestern waters, have a significantly higher probability of occurrence in southwestern waters than in other areas. Another example is Traj (*Trachurus japonicus*). Although Traj is the main target of net fishing in northeastern waters, its larvae and juveniles were mainly found on the northwestern shelf, which serves as a nursery area. Figure 3 shows the catch rate of each fish species in the above sea areas.

In addition to spatial probability, the temporal probability of fish occurrence in different seasons was also analyzed. The study results showed that winter and spring had the highest numbers of larval and juvenile fish of a variety of species, and the probability of occurrence in winter and spring for most species was over 50%. As displayed in Figure 4, the larvae and juveniles of Notosudidae spp. (No), *Sigmops gracilis* (Sg), and Mugilidae spp. (Mu) mainly appeared in winter, among which Mu was the most abundant; these species relocated to other areas through ocean currents or migrated in other seasons. We also found that larvae and juveniles of *Decapterus macrosoma* (Dea), *Decapterus maruadsi* (Deb), and *Decapterus* spp. (Dex) were primarily seen in spring, while those of *Trachurus japonicus* (Traj), which are similar in appearance, mostly appeared in winter or summer. *Auxis* spp. (Au), Dea, and Traj were not found in autumn.

### 3.1. Cluster Analysis of Fish Larvae and Juveniles

The composition of fish larvae and juveniles in spring was analyzed over time, and cluster analysis was performed on the composition data. Some of the results regarding the suggested number of clusters are shown in Figure 5. The model suggested that the stations should be divided into three groups (Figure 6). After the number of clusters was set to three, it was found that the three station clusters exhibited two-dimensional variation, with the first principal component accounting for 22.5% of the variation and the second principal component accounting for 9.9%. By overlaying the results on the map, we found that the first group (red) was mainly distributed in the northwestern continental shelf area, the second group was distributed in the area outside the continental shelf, and the third group was scattered in the area outside the continental shelf. The results showed that the primary principal component, which grouped stations based on different compositions of fish larvae and juveniles represented the difference between areas on and off the continental shelf. In addition, there was a sporadic third group of stations in the noncontinental shelf area. These stations were found in the offshore area of the eastern sea and at the 26th station, areas through which the Kuroshio Current or its tributaries passed (Figure 7).

### 3.2. Temporal and Spatial Variation in Fish Larval and Juvenile Abundances in Different Depth Zones

The study revealed significant spatial differences in the abundance of larval and juvenile pelagic fish. The abundance of fish larvae and juveniles was higher from the northwestern (Figure 1b area 1) shelf area to the Penghu island region (Figure 1b area 6), and the southwestern (Figure 1b area 5) and northeastern (Figure 1b area 2) seas also exhibited higher abundances. Regarding fish species composition, there were higher proportions of larval and juvenile pelagic fish at several stations in the northwest (Figure 1b area 1) and southeast (Figure 1b area 3), while in other areas, species from all three depth zones exhibited relatively even proportions. Stations near Green Island had a higher proportion of mesopelagic larval and juvenile fish species (Figure 8).

According to the time-series variation analysis, the abundances of fish larvae and juveniles in these three different depth zones were relatively even from 2007 to 2008, ranging from 2.1 to 3.5. Since 2009, the abundances of species in all three layers have increased significantly, with pelagic, mesopelagic, and demersal abundances of 9.5, 4.4, and 7.0, respectively, which are far greater than the abundances in the previous two years. In 2009, pelagic species increased to become the most abundant in all depth zones. In 2012, 2015, and 2017, the abundance of pelagic species exceeded 10, remaining high. In contrast, the demersal zone had the lowest species abundance among the three zones from 2010 to 2015, and only in 2016 did the abundance in this zone surpass that in the mesopelagic zone to rank second (Figure 9).

## 4. Discussion

Taiwan is located at the intersection of the tropics and subtropics of the western Pacific Ocean and is rich in biodiversity; here, 298 fish families have been discovered [58]. The fish larvae and juveniles captured in this study included 161 families, of which the 38 families with the highest abundance were representative and used in the final analysis. The families of the 38 most common species covered in this study significantly contribute to Taiwan’s fisheries economy. Their early-life abundance is closely related to the recruitment of fishery resources and requires continuous monitoring by scientists [53].

This study showed that fish larvae and juveniles were highly abundant in the waters of northwestern Taiwan (Figure 8). Past studies have also found a high abundance of fish larvae and juveniles, including anchovy stocks (*Engraulis japonicus*, *Encrasicholina punctifer*, and *Encrasicholina heteroloba*), near the estuary in this region, resulting in the establishment of a local anchovy fishery [59]. Anchovy fisheries result in bycatch of some economic fish larvae and juveniles in summer. Therefore, the local government has expressly stipulated that the anchovy fishery remain closed from 15 June to 14 September every year to avoid mixed catches. This management direction highlights the importance of spatial (Figure 3) and seasonal (Figure 4) studies for endemic fisheries research. For example, if a fish stock is overfished and needs protection, samples such as those collected in this study can clearly reveal the spatiotemporal characteristics of its early life history, which can serve as an essential reference for fisheries management.

In this study, distinct subgroups of fish larval and juvenile community were identified by cluster analysis. The two most important subgroups were located in the continental shelf area and the deep-sea area. The third subgroup was scattered in the deep-sea area. This result may be caused by the following reasons. First, different ecosystems have different biodiversity levels, and the services that these ecosystems provide also differ [60]. Similarly, fish reproduction is naturally variable, and the selection of spawning areas differs according to different evolutionary pressures in individual fish species [39,40,42]. Second, the predators and prey in the two regions differ. In addition, the environmental variables, such as currents, water quality, and temperature changes, were very different. The probability of the survival of individual fish species may drive the differences in the abundance and diversity of larvae and juveniles in the two locations [44]. The third cluster in this study was sporadically distributed throughout the nonshelf area, and there was not much correlation between stations. However, we identified two commonalities. First, these stations were distributed in areas where the Kuroshio Current flows. Second, these stations were farther from land. Additionally, the third cluster was distributed in areas that experience regular monsoons (northeastern and southwestern waters). The main reason for this result may be related to monsoon-driven ocean currents in the waters around Taiwan. For example, a study by Hsieh et al. [61] revealed that some fish species in northeastern waters are driven into the Taiwan Strait by monsoons. Hsieh, Lo, and Wu [11] also found that monsoons could affect the Kuroshio tributaries and cause fish larvae and juveniles to spread from the Pacific Ocean into nutrient-rich areas along the southwestern coast. This phenomenon means that the region is unable to maintain a stable state at any time and is therefore statistically independent, as indicated by its principal components. This study divided the ocean into three regions along the coast of Taiwan; these divisions can be used for further research to explore individual characteristics. If a study includes regions in the third cluster, researchers should consider the possible impact of other factors (e.g., monsoons) on this region.

By analyzing the temporal changes in the abundance of larvae and juveniles at different depths, we found that the abundances of species at the three different depths, which were relatively even before 2008, have undergone dramatic changes since 2009. A significant increase in abundance was not found in previous studies, but the catch of Taiwan’s coastal fisheries reached a new high in 2018, which may be related to our finding. Regarding the possible explanation for the increase in the abundance of pelagic species since 2009, we suspect that abundance is density-dependent.

This inference of density dependence is based on data collected during a natural disaster in Taiwan in 2009. One of the deadliest typhoons in Taiwan’s history, Typhoon Morakot, struck Taiwan in 2009 [62], resulting in a 4-day rainfall total of 2777 mm, strong storm surges, and sea surface waves in the waters around Taiwan [63]. While studies have shown that such tropical depressions significantly affect upwelling enhancement [64], storms have destroyed entire villages due to landslides [65], which also led to the overturn of a large area of coral reefs on the seafloor, with the coral coverage decreasing sharply from 32.72% to 5.39%; other algae filled the places where corals originally grew [66]. Thus, the ecological environment of demersal species was likely significantly affected. Moreover, some of the nutrients may have shifted from the demersal to the pelagic zone.

From 2010 onward, we observed markedly different responses to the disruption in species in the three different zones. Compared to highly migratory pelagic species (such as mackerel and tuna) that have high growth and reproductive rates, demersal fish are mostly sedentary K-selection species and exhibit slow growth, a long lifespan, and a low reproductive rate [67], which may be even more disadvantageous in extreme climates.

Past research has shown that warming affects pelagic fish much more than demersal fish [68] and that demersal fish are more adaptable than pelagic fish to climate change [69]. However, climate changes, including warming, may pose problems not only due to increasing water temperatures but also due to the consequent increase in the probability of extreme climatic events that may impact benthic ecosystems, which merits further attention [20,70,71]. Because demersal fishes are not a large catch, we may not observe stock changes in catch reports. However, our data provide evidence that their resources have changed dramatically in recent years, which may also affect marine ecosystems.

The United Nations attaches great importance to climate issues, and in 2015, 17 specific goals were issued as follow-up development directions. These goals consider the different levels of scientific and technological development in various countries, and countries are encouraged to incorporate these goals into domestic policies and action plans according to their current status. Under this framework, all countries should focus on sustainable economic and social development and strive to obtain effective methods, models, and tools to achieve the Sustainable Development Goals (SDGs) and ensure the sustainability of fisheries in the waters surrounding their country.

## 5. Conclusions

Overall, this study considered the importance of fish larval and juvenile occurrence at different times and in different spaces for fishery resource management. The community structure of fish larvae and juveniles could be described by three principal components, and we found a possible connection between the region of the third cluster and monsoons. Spatiotemporal analysis also revealed that Typhoon Morakot may have had different impacts on species in different depth zones in the surrounding waters of Taiwan and highlighted the vulnerability of demersal species to extreme climate events. The early life history stages of fish are critical determinants of the abundance of fishery resources; larval and juvenile fish and various plankton also play crucial roles in the ecosystem. The consequences of climate change could be profound and can no longer be ignored.

## Figures and Tables

**Figure 1 animals-12-01890-f001:**
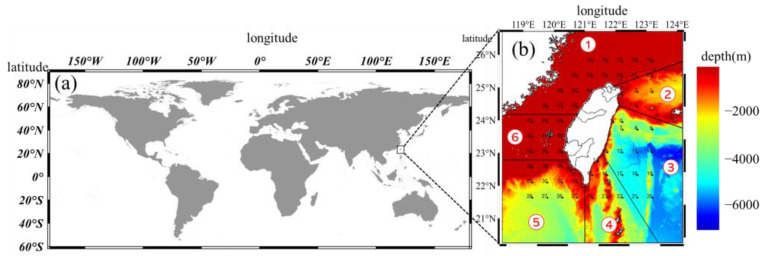
Study areas: (**a**) Taiwan’s location on a world map. (**b**) The study area includes stations (black points) and sea areas (separated by solid black lines and marked with red numbers in white circles) around Taiwan from 2007 to 2019. The black dashed lines indicate contours with depths of 200 m and bathymetry. The areas were defined as follows: the northwestern (area 1), northeastern (area 2), southeastern (area 3), southern (area 4), southwestern (area 5), and Penghu island (area 6) regions.

**Figure 2 animals-12-01890-f002:**
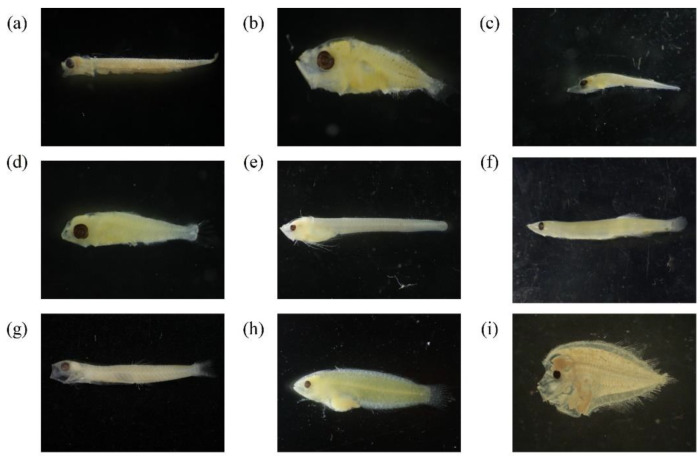
Photographs of the top three catches of fish species (larvae and juveniles) from three different depth zones (pelagic, mesopelagic, and demersal) in this study (around Taiwan from 2007 to 2019): (**a**) *Engraulis japonicus* (top catch in the pelagic zone)*;* (**b**) *Decapterus* spp. (2nd highest catch in the pelagic zone); (**c**) *Trichiurus* spp. (3rd highest catch in the pelagic zone); (**d**) *Diaphus* B group (top catch in the mesopelagic zone); (**e**) *Bregmaceros* spp. (2nd highest catch in the mesopelagic zone); (**f**) *Vinciguerria nimbaria* (3rd highest catch in the mesopelagic zone); (**g**) Gobiidae spp. (top catch in the demersal zone); (**h**) Labridae spp. (2nd highest catch in the demersal zone); and (**i**) Bothidae spp. (3rd highest catch in the demersal zone).

**Figure 3 animals-12-01890-f003:**
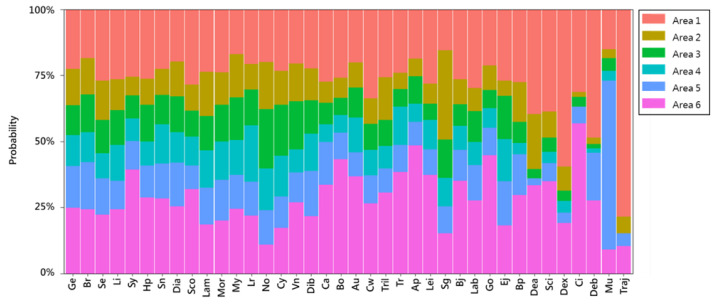
Stacked bar chart of the catch rates of various fish species according to collection area in the waters around Taiwan (2007–2019). The fish codes are defined as follows: *Trachurus japonicus* as Traj, *Engraulis japonicus* as Ej, *Trichiurus* spp. as Tril, *Synagrops* spp. as Sy, *Auxis* spp. as Au, *Decapterus macrosoma* as Dea, Bramidae spp. as Br, Gempylidae spp. as Ge, *Decapterus* spp. as Dex, *Decapterus maruadsi* as Deb, Mugilidae spp. as Mu, Clupeidae spp. as Cl, *Benthosema pterotum* as Bp, *Ceratoscopelus warmingi* as Cw, *Cyclothone alba* as Cy, *Lestrolepis* spp. as Lr, *Vinciguerria nimbaria* as Vn, *Myctophum orientale* as Mor, *Hygophum proximum* as Hp, *Stomias nebulosus* as Sn, *Lampanyctus* spp. as Lam, *Bregmaceros* spp. as Bj, Myctophidae spp. as My, *Diaphus* A group as Dia, *Diaphus* B group as Dib, Serranidae spp. as Se, *Sigmops gracilis* as Sg, *Trachinocephalus myops* as Tr, Notosudidae spp. as No, Apogonidae spp. as Ap, Labridae spp. as Lab, Callionymidae spp. as Ca, *Lestidium* as Li, Bothidae spp. as Bo, Scorpaenidae spp. as Sco, Leiognathidae spp. as Lei, Gobiidae spp. as Go, and Sciaenidae spp. as Sci.

**Figure 4 animals-12-01890-f004:**
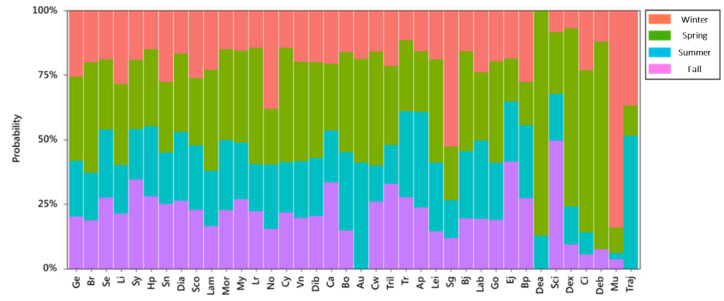
Analytical results of the occurrence probability of larvae and juveniles of individual fish species in different seasons around Taiwan (2007–2019). The fish codes are defined as follows: *Trachurus japonicus* as Traj, *Engraulis japonicus* as Ej, *Trichiurus* spp. as Tril, *Synagrops* spp. as Sy, *Auxis* spp. as Au, *Decapterus macrosoma* as Dea, Bramidae spp. as Br, Gempylidae spp. as Ge, *Decapterus* spp. as Dex, *Decapterus maruadsi* as Deb, Mugilidae spp. as Mu, Clupeidae spp. as Cl, *Benthosema pterotum* as Bp, *Ceratoscopelus warmingi* as Cw, *Cyclothone alba* as Cy, *Lestrolepis* spp. as Lr, *Vinciguerria nimbaria* as Vn, *Myctophum orientale* as Mor, *Hygophum proximum* as Hp, *Stomias nebulosus* as Sn, *Lampanyctus* spp. as Lam, *Bregmaceros* spp. as Bj, Myctophidae spp. as My, *Diaphus* A group as Dia, *Diaphus* B group as Dib, Serranidae spp. as Se, *Sigmops gracilis* as Sg, *Trachinocephalus myops* as Tr, Notosudidae spp. as No, Apogonidae spp. as Ap, Labridae spp. as Lab, Callionymidae spp. as Ca, *Lestidium* as Li, Bothidae spp. as Bo, Scorpaenidae spp. as Sco, Leiognathidae spp. as Lei, Gobiidae spp. as Go, and Sciaenidae spp. as Sci.

**Figure 5 animals-12-01890-f005:**
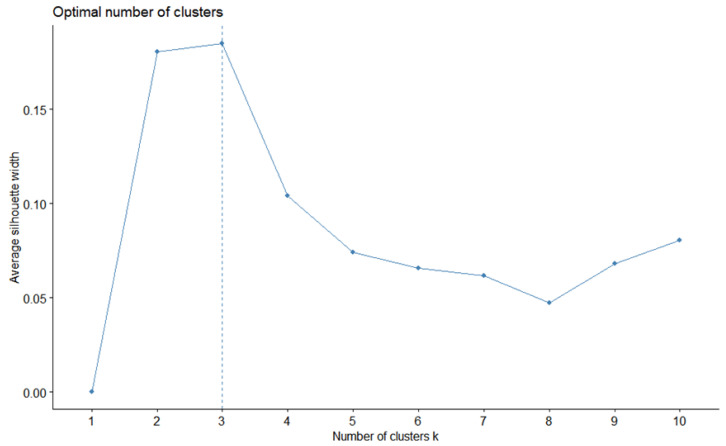
The optimal number of clusters for the analysis of fish larvae and juveniles in the waters around Taiwan from 2007 to 2019. The dotted line indicates the optimal number of clusters, which is located at the point where the number of clusters (x-axis) corresponds to the largest average silhouette width (y-axis).

**Figure 6 animals-12-01890-f006:**
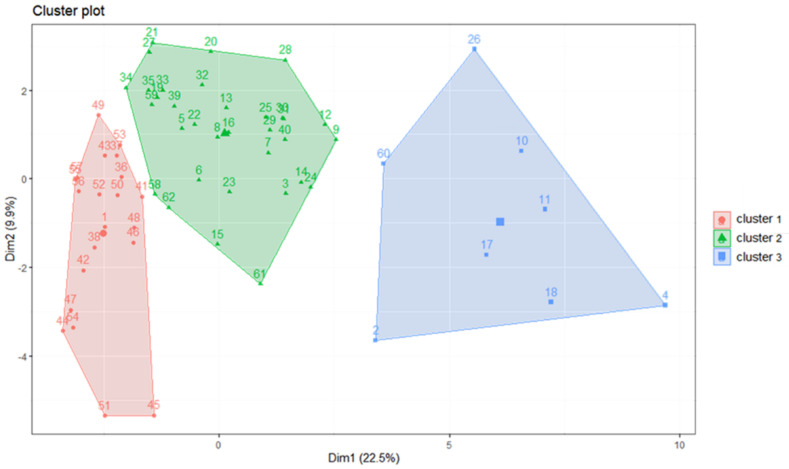
Cluster plot of fish larvae and juveniles collected around Taiwan from 2007 to 2019.

**Figure 7 animals-12-01890-f007:**
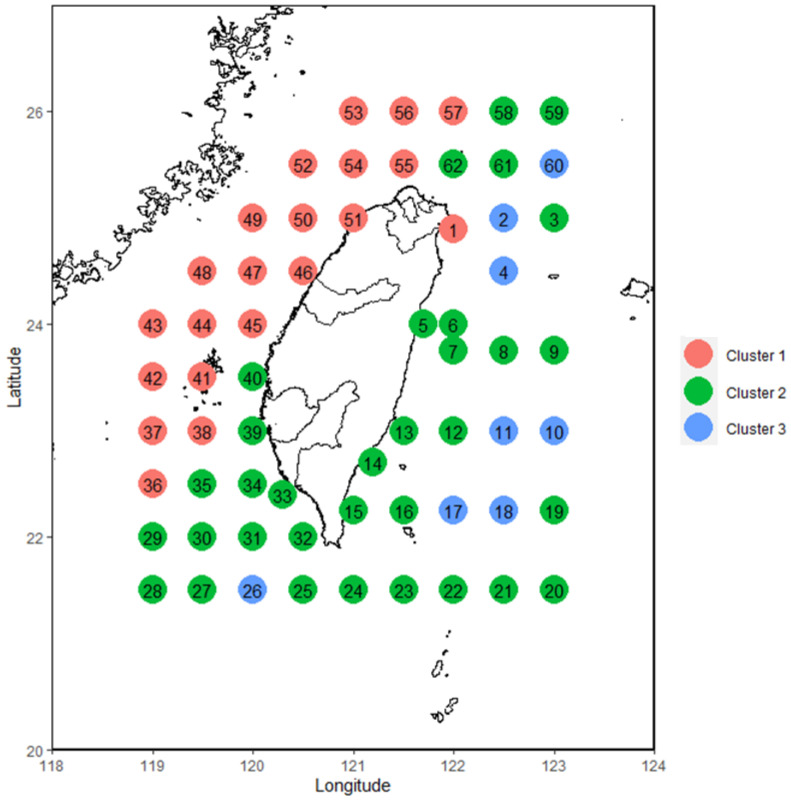
Geographic distribution of the cluster analysis results of fish larvae and juveniles collected around Taiwan from 2007 to 2019 overlaid on a map.

**Figure 8 animals-12-01890-f008:**
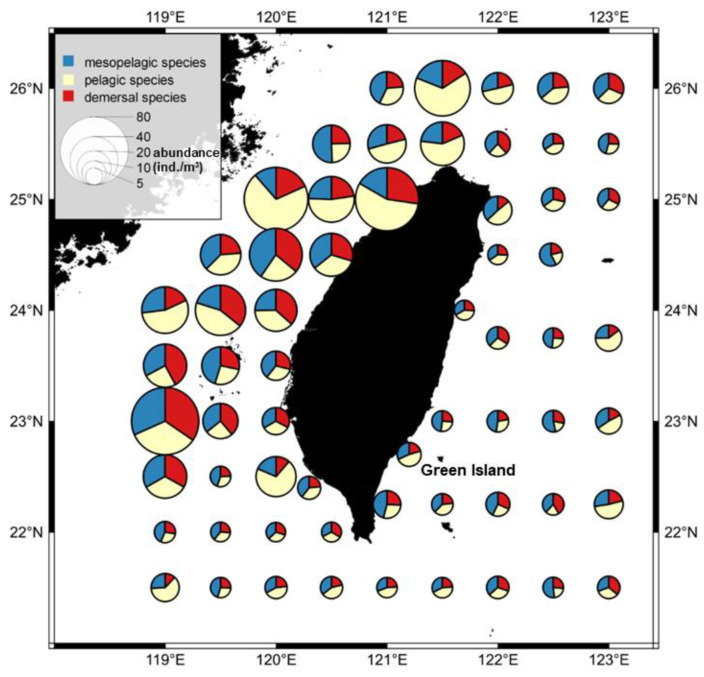
Spatial distribution of the abundance of fish larvae and juveniles in the pelagic, mesopelagic, and demersal zones in the waters around Taiwan between 2007 and 2019.

**Figure 9 animals-12-01890-f009:**
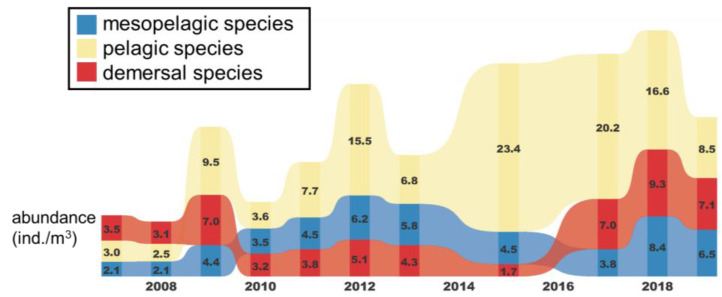
Results of time series analysis of the abundance of fish larvae and juveniles in the pelagic, mesopelagic, and demersal depth zones in the waters around Taiwan from 2007 to 2019.

**Table 1 animals-12-01890-t001:** Codes and habitat depth zones of 38 major species of fish (larvae and juveniles) in the waters around Taiwan from 2007 to 2019. The depth zones of different fish species are listed according to the classification methods of Sassa and Konishi [25] and Chen, Chen, Wang and Lee [13].

No	Family	Species	Code	Depth Zone
1	Carangidae	*Trachurus japonicus*	Traj	pelagic
2	Engraulidae	*Engraulis japonicus*	Ej	pelagic
3	Trichiuridae	*Trichiurus* spp.	Tril	pelagic
4	Acropomatidae	*Synagrops* spp.	Sy	pelagic
5	Scombridae	*Auxis* spp.	Au	pelagic
6	Carangidae	*Decapterus macrosoma*	Dea	pelagic
7	Bramidae	spp.	Br	pelagic
8	Gempylidae	spp.	Ge	pelagic
9	Carangidae	*Decapterus* spp.	Dex	pelagic
10	Carangidae	*Decapterus maruadsi*	Deb	pelagic
11	Mugilidae	spp.	Mu	pelagic
12	Clupeidae	spp.	Cl	pelagic
13	Myctophidae	*Benthosema pterotum*	Bp	mesopelagic
14	Myctophidae	*Ceratoscopelus warmingi*	Cw	mesopelagic
15	Gonostomatidae	*Cyclothone alba*	Cy	mesopelagic
16	Paralepididae	*Lestrolepis* spp.	Lr	mesopelagic
17	Gonostomatidae	*Vinciguerria nimbaria*	Vn	mesopelagic
18	Myctophidae	*Myctophum orientale*	Mor	mesopelagic
19	Myctophidae	*Hygophum proximum*	Hp	mesopelagic
20	Stomiidae	*Stomias nebulosus*	Sn	mesopelagic
21	Myctophidae	*Lampanyctus* spp.	Lam	mesopelagic
22	Bregmacerotidae	*Bregmaceros* spp.	Bj	mesopelagic
23	Myctophidae	spp.	My	mesopelagic
24	Myctophidae	*Diaphus* A group	Dia	mesopelagic
25	Myctophidae	*Diaphus* B group	Dib	mesopelagic
26	Serranidae	spp.	Se	mesopelagic
27	Gonostomatidae	*Sigmops gracilis*	Sg	mesopelagic
28	Synodontidae	*Trachinocephalus myops*	Tr	demersal
29	Notosudidae	spp.	No	demersal
30	Apogonidae	spp.	Ap	demersal
31	Labridae	spp.	Lab	demersal
32	Callionymidae	spp.	Ca	demersal
33	Paralepididae	*Lestidium*	Li	demersal
34	Bothidae	spp.	Bo	demersal
35	Scorpaenidae	spp.	Sco	demersal
36	Leiognathidae	spp.	Lei	demersal
37	Gobiidae	spp.	Go	demersal
38	Sciaenidae	spp.	Sci	demersal

## Data Availability

All data are contained within the article.

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
