# Peer review of "Spatiotemporal Characteristics of Fish Larvae and Juveniles in the Waters around Taiwan from 2007 to 2019"

_animals, 2022, doi:10.3390/ani12151890_

Round 1
Reviewer 1 Report
Animals-1750661: Spatiotemporal characteristics of fish larvae and juveniles in the waters around Taiwan from 2007 to 2019
The authors did a great effort to improve the manuscript. Congratulations for that. However, some points still need consideration before publication.
Specific comments:
Line 137: should be “The fish larval and juvenile abundance”.
Lines 330-333: the authors repeated a lot the word “clusters”. Can you change a little this?
Lines 403-404: the authors did not correct this sentence according to the response to my previous comment (In my opinion, the authors cannot be so direct in this affirmation, maybe add “can be”).
Figure 1 caption: it will be good to add the period of the study “between 2007 to 2019”, as I suggested before.
Figure 2 caption: it will be good to add the location and the period of the study “around Taiwan from 2007 to 2019”, as I suggested before.
Reviewer 2 Report
The authors need to describe the vessel they used.
What is an ORI net, please describe.
Regarding net fishing, so you maintained vessel speed not wire angle? Maintaining wire angle is the norm for oblique bongo tows.
Please describe how you define how you differentiate juveniles from larvae.
Formula 164: So this doesn't take into account region? Later in the results it appears it does? If so you need an additional subscript to include region.
What statistical program did you use?
Formula 192: Are these done in UTM or raw Lat Long? Raw lat/long has many errors that need to be addressed.
Line 204: There is a very large gap in the methods. I could not fully review this manuscript without these.
Line 298: Any area referenced in the manuscript needs an accompanying figure.
Where are these timeseries coming from?
Author Response
Please see the attachment.

This manuscript is a resubmission of an earlier submission. The following is a list of the peer review reports and author responses from that submission.
Round 1
Reviewer 1 Report
I have completed the review of manuscript “Spatiotemporal characteristics of fish larvae and juveniles in the waters around Taiwan from 2007 to 2019” by Yen et al. The authors analyzed the larval and juvenile fish data from 2007–2019 field surveys around Taiwan, and they found three clusters of larval and juvenile fishes in spring, and the increase in abundance of pelagic species. This manuscript is generally well written, and I recommend considering its publication after revisions. Please find my comments below. General comments- Since I did not understand some parts of the manuscript, professional language editing may be useful.
- Use “fish larvae and juveniles” instead of “larvae and juveniles” in important parts of this paper (e.g. goals).
- The analysis part of “Materials and methods” needs to be rewritten (provide more details).
- The meaning of “average” in this manuscript is not very clear (e.g. L275, 281).
- The discussions on the relationships between the environmental factors and fish abundance/compositions should be minimized since this study did not analyze environmental data (thus no scientific evidence).
- A deeper discussion on the increased density in pelagic species since 2009 may be important in this manuscript
Reviewer 2 Report
Animals-1663995: Spatiotemporal characteristics of fish larvae and juveniles in the waters around Taiwan from 2007 to 2019
In their manuscript, "Spatiotemporal characteristics of fish larvae and juveniles in the waters around Taiwan from 2007 to 2019", the authors showed the variability of fish larvae and juveniles in Taiwan for a period between 2007 to 2019.
I believe that this study gives important information regarding the seasonality and spatiality of fish larvae and juveniles around Taiwan, showing that continental shelf and offshore distance were key factors for its variability. However, in my opinion, changes should be considered before publication to improve the manuscript concerning the writing. An extensive revision of the English language by a native speaker is needed. Some parts of the text are confused and some sentences are unfinished. The introduction is a little confused and must be rewritten from general to particular. Some figures are unformatted, and figures and tables captions must be completed. Also, it is important to specify the dates of samples, “from 2007 to 2019” is not enough, more detail about that is needed. Thus, this manuscript could not be accepted for publication yet. I believe authors can perform a better article and, therefore, resubmit a higher quality version. I hope the authors consider my comments and suggestions.
Specific comments:
Abstract
Lines 9-11: this sentence is confusing, please clarify.
Lines 9, 12 and 13: should be “fish larvae and juveniles”.
Keywords
The keywords “larvae” and “juveniles” are repeated in the title. Please replace them. Keywords must be different from the title, in order to increase the work visibility.
Introduction
The introduction must be rewritten, from the general to particular. A connection between subjects/ideas is needed.
Lines 21-30: in my opinion, this paragraph is not adequate for this introduction, there is no connection with the remaining introduction. Maybe the authors can put it in the methods section.
Lines 38, 40, 43, 45, 46, 48, 90, 95, 98, 100, 103, 104, 108, 111, 114, 116, 118, 122, 124 and 127: the authors must specify the type of larvae when they referred to “larvae” can be from different animal groups. Thus, should be “fish larvae and juveniles”.
Lines 39-41: Please clarify this sentence.
Lines 51-60: this paragraph has no connection with the previous and next paragraphs.
Line 68: should be “When prey are insufficient…”.
Lines 96-97: I do not understand this sentence “it is necessary to maintain a suitable living habitat and prevent fish are not displaced by ocean currents”, please clarify.
Lines 126-127: I do not understand this goal, please clarify.
Material and Methods
In this section, the authors must give more information about the sampling periods, “from 2007 to 2019” is not enough. How many times per year? In each season? Every year? Which months? Etc.
Lines 131, 152, 157, 167, 178, 180, 189, 191 and 199: the authors must specify the type of larvae when they referred to “larvae” can be from different animal groups. Thus, should be “fish larvae and juveniles”.
Line 149: why did the authors divide the samples into two subsamples? How many individuals do you have per sample?
Line 153: Just “juvenile” density? Not fish larvae also?
Lines 165-167: it will be easier to understand if the authors connect the names of sea areas with the numbers in Figure 1. And, of course, add a reference to Figure 1 here.
Lines 202-203: the “t” is not represented in the formula.
Results
Lines 206, 208, 209, 230, 231, 245, 255, 268, 270, 275 and 280: the authors must specify the type of larvae when they referred to “larvae” can be from different animal groups. Thus, should be “fish larvae and juveniles”.
Lines 208-209: incomplete sentence.
Line 214: please add also the scientific name of “mullet”.
Line 216: should be “Traj (Trachurus japonicus)”. It is the first time that it is referred to in the text, thus should has the scientific name.
Line 230: should be “The results showed…”.
Line 237: should be “Decapterus spp.”.
Line 273: “pelagic species and juveniles”? what is it meaning?
Discussion
The discussion can be improved, at least with the connection of contents.
Lines 296, 301, 303, 322 and 340: the authors must specify the type of larvae when they referred to “larvae” can be from different animal groups. Thus, should be “fish larvae and juveniles”.
Lines 301-302: please clarify this sentence.
Lines 304-307: please clarify these sentences. I did not understand their meaning.
Line 305: please add also the scientific name of “anchovy”.
Conclusions
Line 380: should be “fish larvae and juveniles”.
Line 385: should be “The early life history stages are critical…”.
Lines 386-387: In my opinion, the authors cannot be so direct in this affirmation, maybe add “can be”.
Tables and Figures
Figure 1: This figure is unformatted; the photographs are not aligned. It will also be interesting to add a more regional map, showing the location of Taiwan at a global scale (like Fgure 1A), and the present Figure 1, will be the Figure 1B.
Figure 2: This figure is unformatted. Please add also the letters “a”, “b”, etc. to each photograph.
Captions
The captions must be the more complete possible for the reader to understand the table or figure without reading the manuscript.
Table 1: should be “fish larvae and juveniles”. Was this table a compilation of previous studies? If yes, please add the studies to the caption.
Figure 1: Please complete the caption, for example: “Distribution of stations (black points), sea area (separated by blue lines, and marked with red numbers in white circles) and bathymetry in Taiwan between 2007 and 2019”. And please add the name of each area that corresponds to each red number.
Figure 2: Please complete the caption, for example: “Photographs of the top three species of fish larvae and juveniles collected in three different depth layers (pelagic, mesopelagic and demersal) around Taiwan from 2007 to 2019: (a)…”.
Figure 3: Please complete the caption, for example: “Stacked bar chart of the catch rates of various fish species by sea area collected around Taiwan (2007-2019). Legend: …”. Please add also the meaning of each species code, and area.
Figure 4: correct the number of the figure. Please complete the caption, for example: “Analytical results of the occurrence probability of larvae and juveniles of individual fish species in different seasons, around Taiwan (2007-2019). Legend: …”. Please add also the meaning of each species cod, and season.
Figure 5: Please complete the caption. The present caption can be used in random studies, the captions must have all the information for the reader to understand the study.
Figure 6: Please complete the caption. The present caption can be used in random studies, the captions must have all the information for the reader to understand the study.
Figure 7: Please complete the caption, for example: “Geographic distribution map of results from cluster analysis, of fish larvae and juveniles collected around Taiwan (2007-2019).”.
Figure 8: Please complete the caption, for example: “Spatial distribution of the abundance of fish larvae and juveniles at the pelagic, mesopelagic, and demersal depths, around Taiwan between 2007 and 2019.”.
Figure 9: Please complete the caption, for example: “Results of time series analysis of the abundance of fish larvae and juveniles in the pelagic, mesopelagic, and demersal depths, around Taiwan from 2007 to 2019.”.
References
The references are not in accordance with the guidelines of the journal and are confused in along the text. Guidelines (https://www.mdpi.com/journal/animals/instructions#preparation):
- “References must be numbered in order of appearance in the text (including table captions and figure legends) and listed individually at the end of the manuscript.”
- “In the text, reference numbers should be placed in square brackets [ ], and placed before the punctuation; for example [1], [1–3] or [1,3].”
